# A Novel Grammar-Based Approach for Patients’ Symptom and Disease Diagnosis Information Dissemination to Maintain Confidentiality and Information Integrity

**DOI:** 10.3390/bioengineering11121265

**Published:** 2024-12-13

**Authors:** Sanjay Nag, Nabanita Basu, Payal Bose, Samir Kumar Bandyopadhyay

**Affiliations:** 1Department of Computer Science and Engineering, Swami Vivekananda University, Barrackpore, Kolkata 7000121, India; sanjayn@svu.ac.in (S.N.); bosepayal91@gmail.com (P.B.); 2Department of Applied Sciences, Northumbria University, Newcastle NE1 8ST, UK; 3The Bhawanipur Education Society, Kolkata 700020, India

**Keywords:** influenza, coronavirus, symptoms, disease prediction, context-free grammar, Chomsky normal form, syntactic pattern analysis

## Abstract

Disease prediction using computer-based methods is now an established area of research. The importance of technological intervention is necessary for the better management of disease, as well as to optimize use of limited resources. Various AI-based methods for disease prediction have been documented in the literature. Validated AI-based systems support diagnoses and decision making by doctors/medical practitioners. The resource-efficient dissemination of the symptoms identified and the diagnoses undertaken is the requirement of the present-day scenario to support paperless, yet seamless, information sharing. The representation of symptoms using grammar provides a novel way for the resource-efficient encoding of disease diagnoses. Initially, symptoms are represented as strings, and, in terms of grammar, this is called a sentence. Moreover, the conversion of the generated string containing the symptoms and the diagnostic outcome to a QR code post encryption makes it portable. The code can be stored in a mobile application, in a secure manner, and can be scanned wherever required, universally. The patient can carry the medical condition and the diagnosis in the form of the QR code for medical consultations. This research work presents a case study based on two diseases, influenza and coronavirus, to highlight the proposed methodology. Both diseases have some common and overlapping symptoms. The proposed system can be implemented for any kind of disease detection, including clinical and diagnostic imaging.

## 1. Introduction

Disease is a condition that impairs the normal living of an individual. According to Merriam-Webster, a disease is defined as “a condition of the living animal or plant body or of one of its parts that impairs normal functioning and is typically manifested by distinguishing signs and symptoms” [1]. According to the preamble of the World Health Organization (WHO), health is defined as “a state of complete physical, mental and social well-being, not merely the absence of disease or infirmity” [2]. So, any impairment to health can be considered a disease condition. For the treatment of or to seek a cure for a disease, normally, medicine is used. According to the National Cancer Institute (NCI) dictionary, medicine “refers to the practices and procedures used for the prevention, treatment, or relief of symptoms of diseases or abnormal conditions. This term may also refer to a legal drug used for the same purpose” [3]. Medical research is an important field of science and technology that contributes towards a disease’s diagnosis, prevention, and cure. Medical research is an inter-disciplinary research area that includes different branches of science, namely, engineering, medical technology, pharmacology, and toxicology, for the development of medical tools for the detection, diagnosis, and cure of disease, encompassing the development of new medical procedures and medicines for the treatment of disease. Medical research also includes pre-clinical and clinical research to establish new medicines and medical processes so that they are effective and do not cause any harm to an individual.

Large amounts of medical data are generated each day. Medical data include clinical data, pathological images/data, radiological imaging data, and other medical examination data. Moreover, patient health records, in the form of digitized electronic health records (EHRs) or electronic medical records (EMRs), constitute medical data that require efficient storage and subsequent analyses for research. The analysis of an individual patient’s data can be used for a disease’s diagnosis and for general health monitoring. Collective data analyses can be used to predict a disease’s characteristics or trends in particular diseases over a certain geographical area. A good example of such a disease is COVID-19. The large volume of data available provides the opportunity to implement machine learning (ML), algorithm-based solutions [4]. Health-related research is a critical area of science and technology as it deals with the lives and well-being of people. There are several lifestyle-related disorders. Certain diseases, like seasonal diseases, infectious diseases, genetic disorders, pathogenic diseases (caused by fungi, bacteria, viruses, or worms), congenital disorders, cancers, physiological, psychological disorders, and organ disorders ( of the lung, kidney heart, liver, or skin), need to be diagnosed early for a better disease prognosis and healthy living. In this research article, two diseases that have similar symptoms, yet have completely different outcomes/prognoses, are considered. Influenza is not a fatal disease; however, a coronavirus outbreak resulted in a pandemic with millions of casualties all over the world.

The most common diseases that are encountered worldwide are Allergies, Colds, and flu (the short word form of the disease influenza) [5]. Flu is a viral disease, affecting the respiratory tract. It is a highly infectious, airborne disease that is known throughout the globe and is transmitted through sneezing and coughing. This seasonal disease affects all age groups, with minor to severe symptoms. The severity of the condition can directly be linked to the immunological condition of the patient. Though common symptoms are a febrile condition, mostly of a low grade, a lack of immediate treatment can result in a high-grade fever. The disease may progress to primary influenza viral pneumonia, or there may be a further, secondary bacterial infection causing secondary Staphylococcus/Streptococcus pneumonia [6]. The disease detection process is mostly carried out by the medical practitioners from the basic symptoms. However, the disease has overlapping symptoms with other viral infections, like dengue, swine flu, COVID-19, and other fatal viral diseases. To differentiate influenza from such severe types of viral diseases, diagnostic tests are conducted to identify the type of the virus so that the particular disease can be determined. Such tests may include a Rapid antigen test, Molecular Assay, RT PCR, and a virus culture [7].

At least three major pandemics caused by viruses were reported in the last century [8]. One of these pandemics was caused by the influenza A (H1N1) virus. There was a sharp decline noted in the number of cases of seasonal influenza in 2020–2021, during the SARS-CoV-2 pandemic. However, influenza came back in full throttle in 2021–2022 [7]. The outbreak of pneumonia in conjunction with an unknown viral fever was found to propagate easily among the people of Hubei Province in China in late 2019. The disease was spreading fast, with the virulent parasite actively propagating itself and affecting humans worldwide. After thorough scientific studies, it was labelled as novel “Severe Acute Respiratory Syndrome Coronavirus 2” (SARS-CoV-2) or locally as “Coronavirus-19” (COVID-19) [9,10]. The disease became a pandemic and soon spread to almost all the corners of the earth. The outbreak of this pandemic affected the entire world and exposed that a viral disease can bring the entire world to an economic standstill for a couple of years. According to the report by the United Nations Conference on Trade and Development (UNCTAD) in June 2021, there was a loss of USD 4 trillion in Tourism alone [11]. The disease exposed the vulnerability of the present health infrastructure, which was incapable to cope with millions of people getting infected each day. Moreover, unavailable or an inadequate medical infrastructure (even in economically and technologically advanced countries) was the primary reason for high mortality all across the world. The medical infrastructure was stretched in almost all countries, yet a lot of patients died due to a lack of medical facilities.

SARS-CoV-2 was preceded by earlier epidemics, SARS-CoV, which was reported from southern China in 2002, and a similar disease, MERS-CoV, in 2012, reported from Saudi Arabia. All the diseases had certain symptoms that were common to other viral infections affecting the upper respiratory tract. The disease manifests approximately 2 weeks after inception and progresses towards “acute respiratory distress syndrome (ARDS)” [12]. The symptoms of the disease vary, depending on the physiological characteristics and overall health conditions of the host. Certain symptoms are asymptomatic and are only revealed by radiologic examination. However, clinical manifestations can be categorized as typical symptoms, atypical symptoms, imaging findings, and other symptoms [12]. Further, symptoms can be broadly categorized into asymptomatic, mild, moderate, severe, and critical [12]. Figure 1 lists out symptoms exhibited by coronavirus patients. Approximately 86.6% of coronavirus patients fall in the age group of 30 to 79 years. The highest mortality rates among coronavirus patients were reported for people aged 80 and over [13]. The study conducted by the authors of the research article [14] in the early period of the coronavirus pandemic indicated that fever, cough, fatigue, and dyspnea were some noteworthy early signs of coronavirus manifestation. The relationship between gender, age, and comorbidity and their effects on the severity of the symptoms associated with coronavirus were studied on a set of patients in Bangladesh in the article [15]. The analysis, based on early coronavirus data, reported that the rate of hospitalization increased with the patient’s age and the presence of comorbidity.

Artificial intelligence can only support a disease diagnosis by a registered medical practitioner but cannot replace the experience and well-informed skill of a practitioner. The proposed method seeks to use such informed disease diagnoses as undertaken by a medical practitioner with the due support of a validated artificial intelligent system and encode the symptomatic and diagnostic information in the form of syntactic grammar. Such an encoded string can be used to define the disease symptoms using well-defined production rules in parity with informed, technologically supported practitioner decision making. Such a string can further determine the probable diagnosis, based on the grammar and defined production rules. The encoded string is in a standardized form that can be further converted to a QR code for transmission to the cloud. The QR code can be identified by a pharmacist, by another medical practitioner, and interpreted uniformly. The patient can get medicines directly from a pharmacist for a minor. The patient might consult a doctor through direct contact or through para-medical personnel, using a remote/tele medicine application.

Medical data are highly confidential in nature. There must be adequate protection for the storage and transmission capabilities of such personalized data. Data in the form of strings can be subjected to abuse by fraudsters and scammers. Such data should be protected by additional encryption layers so that the data can be used by authorized people/organizations, who can use the data meaningfully and responsibly. The distributed storage of data contributes to the additional security of the data by providing layered security at the application level. The local application will be able to protect the local data at the user/patient end. The institutional (hospitals, clinics, and pathological testing centers) data (EMR/EHR) can reside on the local server or cloud servers as per the institutional data storage and protection policies along with application authentication policies introduced by the application. The importance of data storage and protection is paramount to any application that deals with confidential public information. Data sharing must also ensure that the sharing parties are authenticated by the application before any data transmission.

While Section 2 provides a detailed overview of the computer-assisted technology used for disease diagnosis/identification, the use of Artificial Neural Networks (ANNs) for assisting diagnoses and disease estimations by registered medical practitioners has been discussed in detail in this section. After setting out the background for the use of computer-assisted technology in disease diagnosis and recording, Section 3 provides a detailed overview of the proposed methodology for the symptomatic recording of diseases, along with technological support for helping the disease diagnosis and the subsequent secure sharing of such sensitive information among authorized personnel (namely, doctors, hospitals, and pharmacists) in a resource-efficient way, using grammar. Section 4 documents the recording of symptomatic information, along with the disease diagnoses for COVID-19 and influenza and the subsequent sharing of such recorded string information using QR codes. The examples documented are largely reliant on symptom-based diagnoses, but the proposed method is extendable to incorporate patients’ medical history, test results, and algorithm support that can influence medical decision making by registered medical practitioners/doctors.

The novelty of the proposed work lies in the use of the Chomsky normal form to record the basis, the symptoms, the test results, and the algorithm support used by medical practitioners within the nosological framework to facilitate disease diagnoses. The use of such recording of information and the subsequent development of a parser using the Cocke–Younger–Kasami (C-Y-K) technique supports the resource-efficient storage and sharing of sensitive information in a paperless, yet seamless, manner. With the growing influx of pandemics in the last decade, the proposed methodology is timely and economically relevant to the recording and sharing of sensitive information among authorized personnel for easy access to the medical needs of a patient, without them feeling particularly discriminated against.

## 2. Literature Review

The research on disease prediction using computer/statistical-based methods/models is a well-treaded domain. Research papers on the use of general image processing on imaging data are abundant in the literature. However, contemporary research focuses on using image processing to generate features for some ML models to work on. Initially, the use of ML algorithms to classify a new set of features/symptoms as normal or diseased, based on training on a known set of features or symptoms, was adopted. However, with the introduction of Neural Networks (NNs) and Artificial Neural Networks (ANNs), the prediction systems improved significantly. Further improvements in medical diagnostic technology were possible following the implementation of Convolution Neural Networks (CNNs) and Recurrent Neural Networks (RNNs), based on Deep Neural Networks (DNNs) or Deep Learning (DL), in medical data/images [16]. A CNN includes a convolution layer to retrieve local and global features. A DNN has several hidden, connected layers that perform batch normalization and pooling. Contemporary research has abundantly contributed to the disease detection domain.

Several research works used supervised/unsupervised learning algorithms for heart disease detection. The authors of [17] implemented the Naive Bayes (NB) algorithm for predicting heart disease. Similarly, the authors of [18] used Support Vector Machine (SVM) and Random Forest classifiers for detecting heart disease in acute rheumatic fever cases, whereas the research article [19] implemented a host of classifiers like KNN, SVM, NB, Decision Trees, and Logistic Regression. Random Forest and SOM showed good classification results, based on a dataset from the Ministry of National Guard Health Affairs (MNGHA) on the adult population, in a study to detect diabetes mellitus in [20].

The research work [21] applied NNs for diabetes detection, while [22] implemented NNs in the diagnosis of sepsis in children. The research article [23] implemented ResNets for skin lesion classification. Rahman et al. used CNNs for the detection of COVID-19 [24]. The authors Eweje et al. [25] implemented DL methods on a combination of MRI images and clinical features to identify the nature of bone lesions. Authors Fan et al. [26] used DeepFM to predict the recurrence of Cushing’s disease post-surgery from a dataset of 354 patients. The identification of breast masses from mammography was implemented by Jiao et al. [27] using CNNs. Cheng et al. [28] used EHR data and DL for phenotypic analyses. The detection of heart disease/failure was performed by Ning et al. (Congestive Heart Failure using CNNs) [29]. Sajja and Kalluri (heart disease prediction using CNNs) [30] and Lee et al. [31] applied “long short-term memory (LSTM)” DL-RNNs to ECG signals, and Choi et al. [32] used “gated reccurent unit GRU”-based RNNs for heart failure identification. DL, in images obtained from “Endoscopic Ultrasonography (EUS)”, was used by Seven et al. [33] to analyze and predict malignancy in tumors originating from the gastro-intestinal (GI) track. Tsai and Tao [34] implemented CNNs on Colonoscopy images to differentiate colon cancer from rectal cancer.

The detection of diseases of the brain is challenging. There are several research articles that have implemented AI in detecting such diseases. The ANN implemented by Soundarya et al. [35] and the Deep FM model developed by Ronge et al. [36] have been used for detecting Alzheimer’s disease. An ensemble of binary classifiers was developed by Gregorio et al. (2021) that use handwriting and drawing tasks to discriminate Alzheimer’s patients from the non-sufferers [37]. Parkinson’s disease detection that used CNNs on speech data was implemented by the author Gunduz [38]. LSTM-based RNNs for the prognoses of dementia patients were developed by Wang et al. [39]. Acharya et al. [40] and Muhammad et al. [41] implemented CNN models on Electroencephalogram (EEG) signals for disease detection. Hossain et al. [42] used DL algorithms for the detection of Epileptic seizures. Extensive research is still ongoing into brain tumor detection/staging using DL, CNNs, and transfer learning using magnetic resonance (MR) images. Some notable works include publications by Zhang et al. [43], Amin et al. [44] (using transfer learning with the pre-trained networks AlexNet and GoogleNet), and Chelghoum et al. [45] (using nine different networks to train the DL model). Other similar works by Kaur and Gandhi [46] (using CNNs), Rehman et al. [47] (using CNN models with transfer learning), Kokkalla et al. [48] (using ResNet V2), Toğaçar et al. [49] (using the proposed DL model BrainMRNet), and Amin et al. [50] (implementing LSTM) are worth reporting.

The two diseases that are in focus in this research article are influenza and coronavirus infections. Several research articles employed computer-based modeling for disease prediction/forecasting. Most of the research articles on computer-based influenza detection have contributed to predicting or forecasting flu or influenza-like illness (ILI) outbreaks using data obtained from hospitals, social networks, and from different organizations. The articles worked upon data using statistical, ML, and DL models to predict the occurrence and the spatial and seasonal distribution of influenza. Some notable works include Aiken et al. [51], for predicting the outbreak of flu using neural networks, and Yang et al. [52], who performed a surveillance study of influenza symptoms on different hospital patients in different locations using statistical methods. Similarly, Ali and Cowling [53] synthesized data collected from different (online/offline) sources to track, predict, and forecast the occurrence of influenza. The use of DL and ML methods on data collected from the Centre for Disease control (CDC); social media platforms, like Twitter; and online news articles for the predicting/forecasting of flu outbreaks was reported by Wahid et al. [54] and Jang et al. [55]. Data Mining methods, using Twitter data, for influenza occurrence prediction were reported in research articles [56,57,58,59].

The coronavirus pandemic provided a huge scope to data scientista and AI researchers for predicting the different aspects of the disease. Medical, pharmacological, and immunological research was focused on understanding the disease, finding ways to combat the pandemic, and preventing the spread of the disease/fatalities. A large amount of data was generated during the ongoing pandemic. This provided opportunities to apply AI and data science tools to screen for and tackle the disease, predict diagnostic outcomes, and forecast outbreaks. There were several aspects of understanding the disease that the computer/data scientists contributed to that helped medical practitioners to manage the disease. Ardakani et al. [60] and Ozturk et al. [61] used CT or X-ray images, along with clinical and demographic data, for the screening of COVID-19 cases, implementing ML, CNNs, and DL in their research dissertations. Chimmula et al. [62] and Chakraborty and Ghosh [63] worked on demographic data. They implemented DL with LSTM and ARIMA models on time-series data for predicting and forecasting the COVID-19 outbreak/spread. There is a plethora of work on image data obtained from X-ray, Computed Tomography (CT), and magnetic resonance (MR) images for the segmentation of lung images, lung lesion detection, and pneumonia detection. These research works utilized every possible method in ML, CNNs, DL, and RNNs for classifying cases of coronavirus and related abnormalities from the image datasets. The list of works is exhaustive; however, these research works did not use the clinical data and symptoms to identify coronavirus cases, as the features used were often abstract. An overlay visualization of them is shown in Figure 2 (VOSviewer, https://www.vosviewer.com/), depicting the associations of different keywords used in the studies mentioned in this section. The authors and the citations as weights are shown in the overlay visualization map depicting the research groups and the citations received by the works in Figure 3.

The storage of medical data is critical to any application pertaining to the development of systems that will help patients interact with stakeholders. Medical data are private/sensitive data and should be protected so that they are not misused. The application proposed requires medical data to be stored in a repository and the diagnosis to be presented in a QR code. Extensive research has been carried out in this domain. Some of the important contributions in this field are summarized in Table 1. The overlay diagram in Figure 4 shows the association of different authors who have contributed to the research domain pertaining to the storage of confidential medical data.

## 3. Proposed Methodology

The proposed method is an extension for the secure and responsible sharing of symptomatic and diagnostic patient information. The effectiveness of the method lies in its ability to encompass the symptomatic conditions of the patients with the informed, digital technology-supported diagnoses provided by registered medical practitioners without being largely resource intensive. The support of present-day digital technology, coupled with the nosological rules used by medical practitioners for disease diagnosis, can be easily represented with grammar and is the inspiration for this research article. The disease is represented by symptoms and the support of well-validated digital technologies that are used by medical practitioners for the determination of a particular disease. The subsection below discusses the grammar used in the case study.

### 3.1. Syntactic Pattern Analysis

A collection of data may have intrinsic patterns. Such data can be textual, signal, or image data. Identifying such inherent patterns using statistical methods or an algorithm is termed as a pattern analysis. A type of pattern recognition is syntactic pattern recognition [108]. In this approach, every object can be expressed as variable cardinality, collections of symbolic and conceptual properties. This enables the representation of pattern structures, can easily model complex relationships among features, and can represent quantitative vectors of features with fixed dimensions that are employed in statistical classification. If the patterns have a defined structure, it can be employed instead of statistical pattern recognition. A string of symbols from a formal language is one approach to represent such structures. In this scenario, the changes in the hierarchy of classes are represented as separate grammar.

The application of a syntactic pattern analysis for disease recognition is demonstrated in the research dissertation by Bandyopadhyay and Manna [109]. The approach is based on context-free grammar in Chomsky normal form. The authors attempted to analyze ECG signals to determine whether the signal was normal or the patient was suffering from right-bundle branch block, left-bundle branch block, left ventricular hypertrophy, left-anterior hemiblock, or left-atrial hypertrophy. The authors implemented wave recognition, the measurement of ECG waves, axis analysis programs, and the arrhythmia detection program as feeds to the syntactic grammar model. The authors classified patients as healthy or identified the particular heart disease using features generated from the ECG waveform data. The application used grammar with a definite set of production rules on the obtained features for classification. Based on the ECG signal, the normal and diseased waveforms could be labelled as either healthy or diseased in this context, using formal grammar.

### 3.2. Chomsky Normal Form

A formal language [110] contains a set of strings, and each of the strings contains a finite set of symbols. Such strings are governed by a set of complex rules. Formal languages are devised to accomplish specific tasks. A programming language is typically a formal language that is governed by a strict set of syntax so that the compiler can interpret the strings to sequences of machine instruction. A natural language also contains a set of rules; however, their development was natural or evolved through the practice of the language. Though the strings of a formal (programming) language are similar to natural language, a computer fails to comprehend this natural language. Therefore, it is necessary to transform the formal language into a language that a computer is able to understand. This language is called context-free grammar [111]. Context-free grammar has the drawback of having lengthy, intricate rules and being quite difficult. To simplify these complex rules, Noam Chomsky proposed a simplified form, which is called Chomsky normal form (CNF) [112]. In general, context-free grammar, G, is a 4-tuple (N, T, Σ, S), where N is a set of non-terminal symbols, T is a terminal symbol, Σ is a Production Rule, and S is a start symbol. According to Chomsky normal form (CNF), the Production Rules are described as follows
Σ→AB; here A and B = Two Nonterminal symbols
A→a and B→b; Here a and b are two terminal symbols

The fundamental benefit is that Chomsky normal form requires around 2n − 1 steps for the formulation of a string of n characters. As a result, a comprehensive examination of all derivations can be used to establish whether a string is defined by the language. This is the key justification for using CNF in this investigation.

### 3.3. Parse Tree Generation

Parsing [113] is the term used to describe the technique of converting data between different formats. It is a part of the translator that aids in organizing a linear text structure in accordance with the grammar, which is a collection of established rules. A parser producer accepts a string as an input and produces a parsing table using the rules of grammar. Two types of parsing are available, Top-Down parser and Bottom-Up parser Table 2.

The Top-Down parser [114] is also known as a predictive parser. This parser produces a tree-like structure, with the root denoting the initial symbol and the leaf nodes representing the string generated by the grammatical rules.

On the other hand, the Bottom-Up parser [115] is also known as a Shift Reducer parser. This method uses the input symbol as the starting point for parsing and builds the parse tree up to the start symbol by going backwards through the rightmost string derivations.

### 3.4. Proposed Algorithm

An input string is established depending on the primitive sequence of the disease characteristics. This type of string representation describes a disease pattern which is composed with the primitive elements of disease symptoms. The string-generating algorithm (Algorithm 1) is described below.
**Algorithm 1:** String generation algorithm using CNF**Input:** String contains with Symptoms **Output:** Whether member of the grammar the input be a string consisting of n characters: a_1_……a_n_the grammar contains R nonterminal symbols R_1_ … R_r_, with start symbol S.The grammar G contains no rules.w is the string of length n to be parsed.Step 1: If w = ϵ (ϵ = empty string) and S → ϵ is a rule in G. Then, Accept the String else Reject the string Step 2: for i = 1 to n            for each production rule R_i_ = a_i_, place the Non-Terminal Value at cell (i, i)Step 3: for i = 2 to n            for j = 1 to n − i + 1                     j = i + j − 1           for k = i to j − 1                      for each production rule check,if cell (i, k) and (k + 1, j) contains the non-terminals of the production rule                               Replace the left side of the production rule in cell (i, j)Step 4: Check if start symbol present in the cell (1, n)                      Then, Accept the string           else                     Reject the string

### 3.5. Diagnostic Criteria

Identifying a class of influenza or coronavirus based on the specified indicators is the goal of this study. In order to identify the category of the aforementioned diseases, the initial step is to determine a set of grammatical characteristics. These grammatical production rules aid in establishing a connection between the symptoms of the experimental disorders. The following step involves making a number of selections based on the symptoms and the regulations. The symptoms experienced by the patient are contrasted with the list of specific symptoms for the aforementioned diseases, and a string is formed as a result. Table 3 and Table 4 provide an illustration of the coronavirus and influenza criteria for diagnoses in this study.

## 4. Experimental Results, Applications, and Discussions

### 4.1. Result

The important task after generating the string-like structure is to analyze the syntax. The syntax analyzer helps to determine the grammar generated by the language. In addition, by generating the parse table, it aids in demonstrating the correctness of the input string derived from the input grammar. In this proposed method, in order to explain the disease pattern, Chomsky normal form for context-free grammar is employed. Other illness symbols for the preferred symptoms are regarded as non-terminals. The grammatical pattern describing these disease conditions is represented below.

D_Diagnosis_ = D(G)G = {N, T, Σ, S}N = {NORM, I, COV, I_1_, I_2_, I_3_, C_1_, C_2_, C_3_, F, C, B, H, T, R, D, F_1_, V, D_1_, L}T = {f, c, h, b, t, r, d, m, l, v, e}S = {NORM, I, COV}

The production rules Σ are defining as follows:
Σ:NORM → FC|FB|FH|FT|FR|CH|BT|TR|RD|DF_1_ | F_1_ L|LV|VD_1_I_1_ → F NORM|C NORM|B NORM|H NORM|T NORM|R NORMI_2_ → NORM I_1_I_3_ → I_1_ I_2_I → I_2_ I_3_C_1_ → I_3_ I_4_C_2_ → I_4_ C_1_C_3_ → C_1_ C_2_C_4_ → C_2_ C_4_COV → C_3_ C_4_F → f; C → c; B → b; H → h; T → t; R → r; D → d; F_1_ → m; L → l; V → v; D_1_ → e

A complete discussion of the various production rules for the suggested approaches is provided in the preceding paragraph. It is essential to infer whether an unknown string, x, belongs to $D_{Diagnosis}\left(G\right)$ or not. If the unknown string is successfully parsed using the above grammar rules, then it will be proven that the unknown string, x, belongs to $D_{Diagnosis}\left(G\right)$. To implement this parser here, Bottom-Up approach is considered. It is, therefore, necessary to determine whether the ambiguous string falls under one of the established rules. The Chomsky normal form is well recognized for context-free grammar. Therefore, the Cocke–Younger–Kasami (C-Y-K) technique [116,117] is used in this approach to construct the parser.

It is employed to determine whether a specific string may be extracted from the language produced by certain grammar. It is also known as the “membership algorithm”, since it determines whether or not the provided string is a part of the grammar. This parser has the benefit of producing a tabular form.

#### 4.1.1. Application: Case Study 1

Let us consider another unknown string, x, which satisfies the disease symptoms. Let us say an unknown string, *x* = *fchbtr*. The parse table, using the CYK algorithm, is illustrated in Table 5. Examining the first row and the last column, which are circled, helps to determine whether certain diseases are more probable to arise. If the start symbol, S, is present in that location, it means that the unknown disease string is accepted by the grammar and it satisfies the rules.

#### 4.1.2. Application: Case Study 2

Let us consider another unknown string, x, which satisfies the disease symptoms. Let us say an unknown string, *x* = *fchbtrdmlve*. The parse table, using the CYK algorithm, is illustrated in Table 6.

The string, obtained as per the observed symptoms, and the diagnosis, obtained based on the production rules of CNF, are converted to a QR code. This code is readable by any QR code scanner. The QR code is automatically generated once the medical practitioner inputs the symptoms and outcomes from an artificially intelligent system along with their informed diagnosis. Hence, another competent doctor, who might be geographically distant, will be able to interpret the symptoms and the predicted diagnosis and can further suggest a medical evaluation or test for further investigation and the subsequent encoding of the information. A pharmacist can give general medicine based on the symptoms and the diagnosis (Figure 5). Since the production rules are based on an established medical foundation and validated digital technology, the interpreted results and encoding are globally acceptable. It will be easier for the patient to carry the diagnostic outcome on a smartphone and, hence, ensure the portability of the system. Figure 6 shows the QR codes and corresponding strings that can be interpreted through any QR scanner anywhere in the world [Code available in figshare—Link provided in Appendix A].

Visual language grammar has been widely studied to model, parse, and implement graphical notations in computing and software engineering problems. Marriott and Meyer [118] laid down the initial concepts in visual language theory, documenting the importance of symbols and diagrams as a concept of human communication against a sequential (textual and verbal) form of communication. The authors Costagliola et al. [119] implemented frameworks for generating and modeling visual notations, based on eXtended Positional Grammar (XPG) that can be implemented for software development methodologies (similar to CASE tools). The authors also implemented a visual language compiler compiler [120] for the parsing of such visual notation systems. Harel [121] discussed visual formalisms, the use of higraphs-based languages, like state diagrams and entity–relationship diagrams, and documented their significance in human–computer interaction. Parsing methodologies for visual grammar were explored by various authors, including Wittenburg [122,123], using predictive, Earley-style parsing for relational grammar. Similarly, Golin [124] proposed a parsing algorithm for visual languages defined by picture layout grammar, where the grammar syntax and the image were used to generate the expression tree. Authors Costagliola et al. [125,126] studied the parsability of linear positional grammar (pLR) and defined a parser for an extended pLALR (look ahead left right) language that can be generated easily by the Yet Another Compiler Compiler (YACC) tool by Bell Laboratories [127]. Further research was conducted by Minas [128], Rekers, and Schurr [129,130] on diagram editors and graphical language implementation. Minas [128] developed a diagram editor and implemented hypergraphs as an internal diagram model and hypergraph parsers for hypergraph syntactic analyses. Rekers and Schurr [129,130] implemented syntax for visual data editing using visual language syntax and spatial relations graphs. Symbol-relation and relational grammar were documented for graphical languages by Ferrucci et al. [131,132] and Wittenburg and Weitzman [123]. Tools for generating intelligent diagram editors and syntax-aware visual language environments were proposed by Marriott et al. [133,134] and Costagliola et al. [135]. The design and rapid prototyping of visual language environments usinga Model-View-Controller (MVC) framework have also been a focus, as discussed by Zhang et al. [136] and Ferrucci et al. [132]. These works on visual language grammar together helped in the understanding, development, and implementation of visual languages in computational environments.

The grammar of the proposed approach can also be represented in visual notations (Costagliola et al., 2002 [137]; Costagliola and Polese 2000 [138]). Visual notation is a language, in which visual symbols constitute the language’s alphabet. They are designed to accept a set of feasible visual strings composed of these visual symbols. Such visual notations are effective for encoding state diagrams, activity diagrams, and Petri nets (Costagliola et al., 2004 [119]). Such visual sentences that encode components and relationships between such components can be syntactically represented in three ways, namely, attribute-based, relation-based, and linear (Costagliola et al., 2004 [119]). An attribute-based representation of the activity diagram (Figure 7) of the proposed grammar is provided in Table 7.

The proposed grammar can further be extended to incorporate relevant medical test results, AI support system predictions, and past medical histories, based on the diagnosis requirements of different diseases. The nosological rules that doctors use to make diagnoses based on all such information will be encoded in the form of grammar and effectively shared among authorized personnel using QR codes. The proposed extension of the model is represented in Figure 8.

### 4.2. Discussion

The most important aspect of human existence is to live a healthy life. However, there are several factors that contribute to ill health. Apart from environmental factors, there are social factors and economic factors that play a significant role in the health of an individual. There are myriad conditions that may lead to a disease condition. People suffer from congenital disorders, lifestyle disorders, the malfunctioning or degeneration of human organs, the malfunctioning of different bodily functions and systems (nervous, vascular), allergies, poison/chemical-induced illness, and pathogens. Pathogens like viruses, bacteria, fungi, and worms cause diseases, are mostly infectious, and spread through common media like air and water. Infection due to pathogens is a common problem that is unavoidable. Such infections are ever-increasing due to more connectivity, interactions, and communications globally. The need for the easy, responsible, and secure sharing of symptomatic and diagnostic information is prevalent in health applications and is the motivation for this research.

Health infrastructure and the use of ICT in public health has been the focus of research since the outbreak of the coronavirus pandemic. The Literature Review Section has discussed, in detail, the focus of ICT research in the health industry. The need for disease detection systems and the interpretation of various clinical data, symptoms, and vast diagnostic imaging resources is a priority in healthcare. The lack of adequate and skilled human resources have always plagued the healthcare industry. ICT and AI as means to replace/support healthcare workers are important to meet the demands of the healthcare industry. The human factor is important in healthcare as it is directly linked to the well-being of an individual. Innovations that can be supported by minimalist technology (for example, the use of smart mobile phones) and can be utilized by any individual with/without the knowledge of ICT are desirable.

The primary objective of this research is to develop a system where a patient can record the symptoms and a rule-based system (following the well-established medical norms) that incorporates the outcomes of validated digital technology systems to support practitioner decision making. The symptoms and diagnoses, when encoded in the form of a QR code, are easily portable. The code can be scanned by medical practitioners in hospitals, diagnostic centers, and pharmacies. The encoded string will be useful for further investigation and drug delivery. The universal nature of standard medicinal practices will ensure that the encoded symptoms and diagnoses will be inferred upon by anybody in the health industry in a uniform manner. Even if the patient travels across the country or to a distant geographical location, the string can be inferred for symptoms and diagnoses. This will further the process of disease treatment in a timely manner and effectively. The research work by Bandyopadhyay and Manna [61] demonstrated that the grammar rules can be used with ECG signals. The authors have generated production rules based on signal patterns that show abnormality. Applying such rules, the authors have detected heart conditions based on CNF. This shows that the context-free grammar and Chomsky normal form can be implemented with other signal- or image-oriented data. So, production rules can be generated for imaging data like X-ray, CT, and MR images with consultations with radiologists. Diagnostic inferences can then be drawn, based on the generated rules.

Despite the fact that statistical pattern recognition is simple, there are numerous real-world challenges for which it may not always be the best solution. The use of AI and ML is useful in cases where there is sufficient fuzziness in the decision-making process. Statistical and ML algorithms are best suited to learn from previous decisions and generate production rules internally for decision making. It is challenging to estimate patterns that contain structural or relational information as feature vectors. These structural details are used by syntactic patterns to classify, as well as describe, the patterns. A collection of production grammar is inferred for each class of pattern in this analysis as part of the pattern recognition predictor. This makes it possible to convey fundamental structural elements in a variety of concise ways for an infinite number of strings. The primary benefit of this strategy is that grammatical production rules can be rewritten and used repeatedly.

## 5. Conclusions

This research paper investigated an alternative way to support disease prediction using context-free grammar. The ease of the relevantly secure accessibility of sensitive data with limited resources to support medical practitioners for information sharing would help strengthen the present medical infrastructure. The pandemic situation exposed the limitations of the medical infrastructure and its inability to reach the masses in the case of massive outbreaks of diseases. A vast number of studies contributed to the domain of disease prediction, computer-aided diagnosis systems, and medical image analysis systems and are still ongoing. However, the prediction of diseases based on symptoms, clinical investigations, and medical imaging is rule-based. The doctors, medical specialists, and radiologists work on certain production rules that define a particular disease condition and differentiate it from a normal case. Such production rules were developed through research and statistical evidence and are well documented in the medical literature. Even in the case of imaging data (of any modality), the radiologists identify an anomaly, if present, in the image data and report anything that is abnormal in size or shape against what they expect to see. The parameters are well defined for the radiologists/pathologists/doctors and, hence, they can conclude the diagnoses or can request for further investigations in support of their inferences. The parameters are correlated, and combinations provide any definite outcome. This study used two diseases that are contemporarily relevant and demonstrate overlapping symptoms. However, influenza is mostly a harmless febrile condition that is cured easily, but coronavirus is a fatal disease that has a high mortality rate. The use of context-free grammar in the simplified form of Chomsky normal form can be utilized to map the parameters (symptoms) as production rules for the grammar. The production rules are based on a medical diagnosis methodology that is uniformly utilized in the diagnosis of the disease. The inference drawn after parsing is the diagnosis based on the production rules provided. The string consisting of the symptoms (as symbols) and the diagnosis is further converted to a QR code for portability. Though this paper has performed experiments with only two diseases, it can be extended to other diseases. Moreover, any kind of medical data can be used to develop the production rules using a definite medical framework. Such production rules can be used to diagnose diseases and the abnormalities present within the data provided.

## Figures and Tables

**Figure 1 bioengineering-11-01265-f001:**
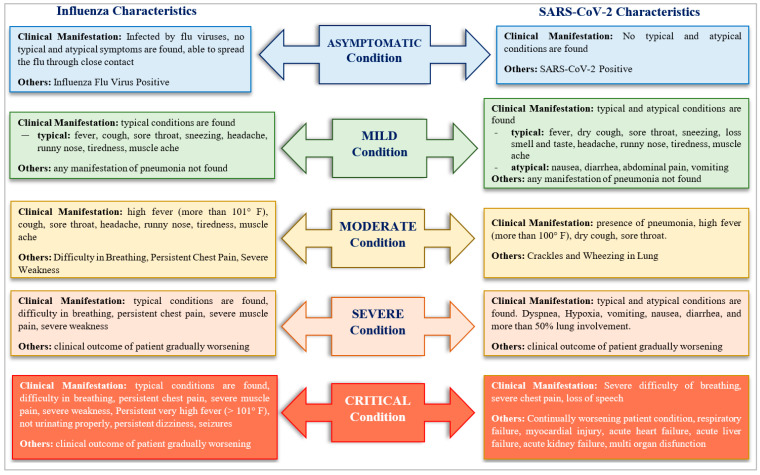
Symptoms of coronavirus infection and influenza for mild to critical cases of the diseases.

**Figure 2 bioengineering-11-01265-f002:**
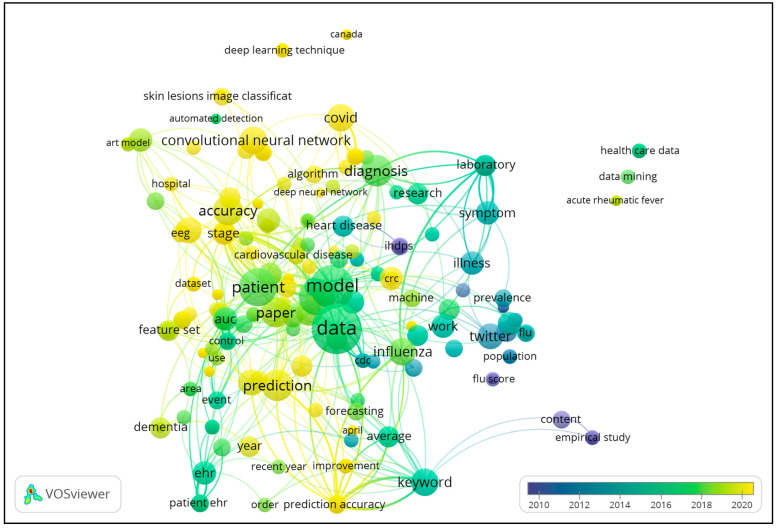
Overlay visualization map of the literature reviewed, showing the association of keywords used and the timeline of the reviewed works.

**Figure 3 bioengineering-11-01265-f003:**
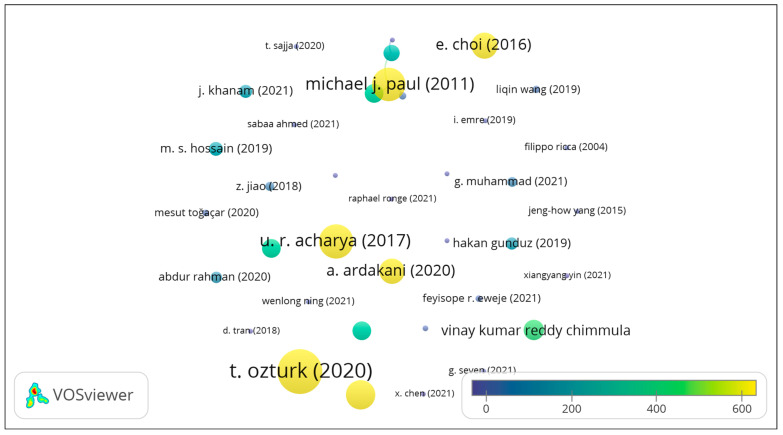
Overlay visualization map of the literature reviewed, showing the authors and the citations of the reviewed works [17,18,19,20,21,22,23,24,25,26,27,28,29,30,31,32,33,34,35,36,37,38,39,40,41,42,43,44,45,46,47,48,49,50,51,52,53,54,55,56,57,58,59,60,61,62,63].

**Figure 4 bioengineering-11-01265-f004:**
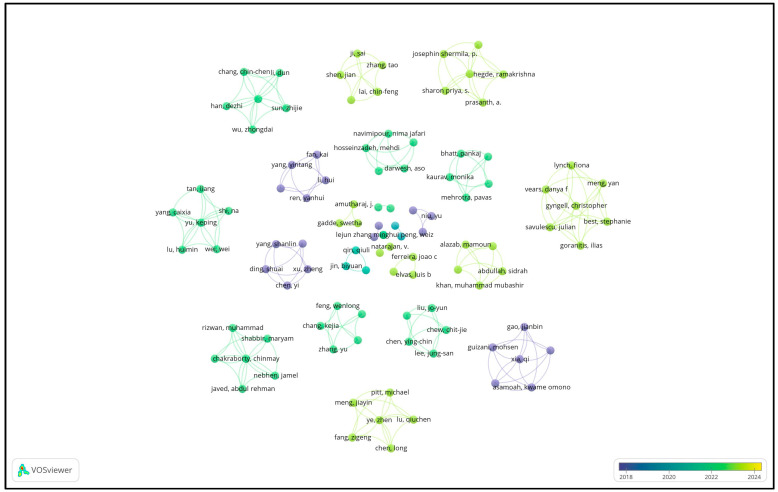
The overlay diagram, showing the authors and the association map of the authors [17,18,19,20,21,22,23,24,25,26,27,28,29,30,31,32,33,34,35,36,37,38,39,40,41,42,43,44,45,46,47,48,49,50,51,52,53,54,55,56,57,58,59,60,61,62,63].

**Figure 5 bioengineering-11-01265-f005:**
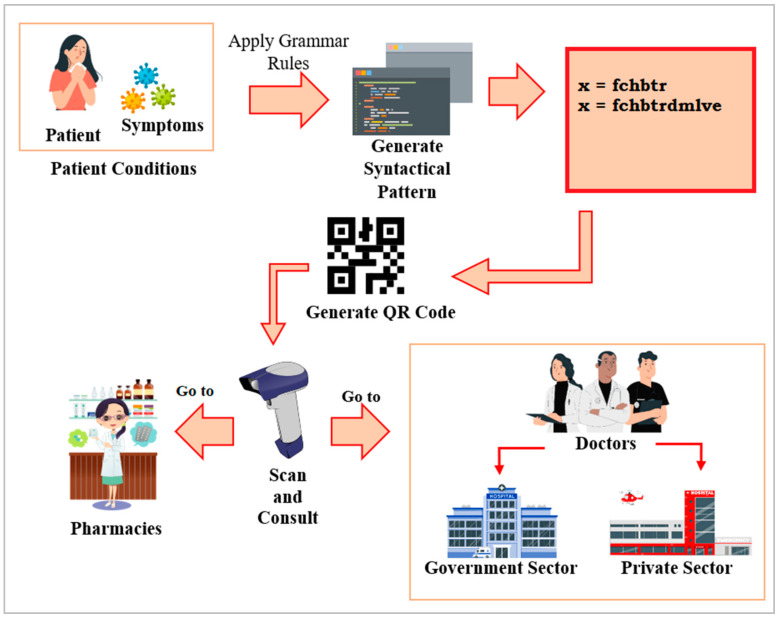
Proposed system of medical infrastructure that involves disease prediction using encoded grammar rules and QR codes for transmission.

**Figure 6 bioengineering-11-01265-f006:**
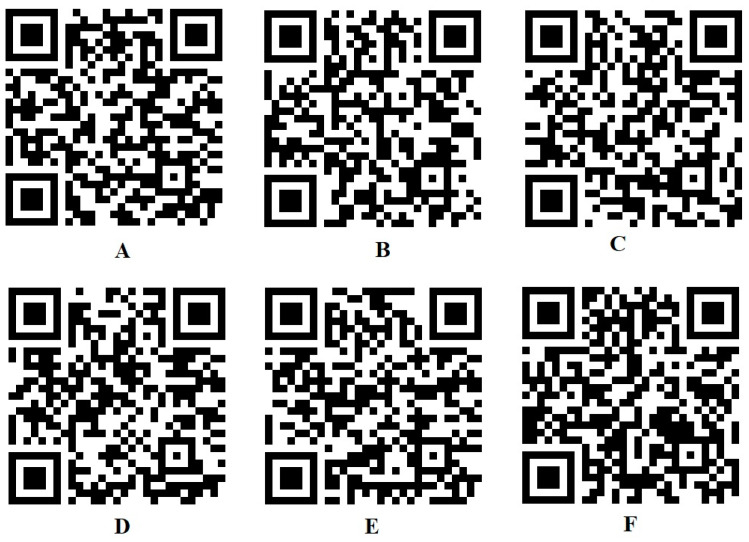
Example of QR code representation for influenza and COVID-19 variations: (**A**) critical COVID-19 (string—fchbtrdmlv); (**B**) critical influenza (string—fchbtr); (**C**) moderate COVID-19 (string—fchtdlv); (**D**) moderate influenza (string—fchbt); (**E**) Severe COVID-19 (string—fchbtdlmv); (**F**) severe influenza (fchtr).

**Figure 7 bioengineering-11-01265-f007:**
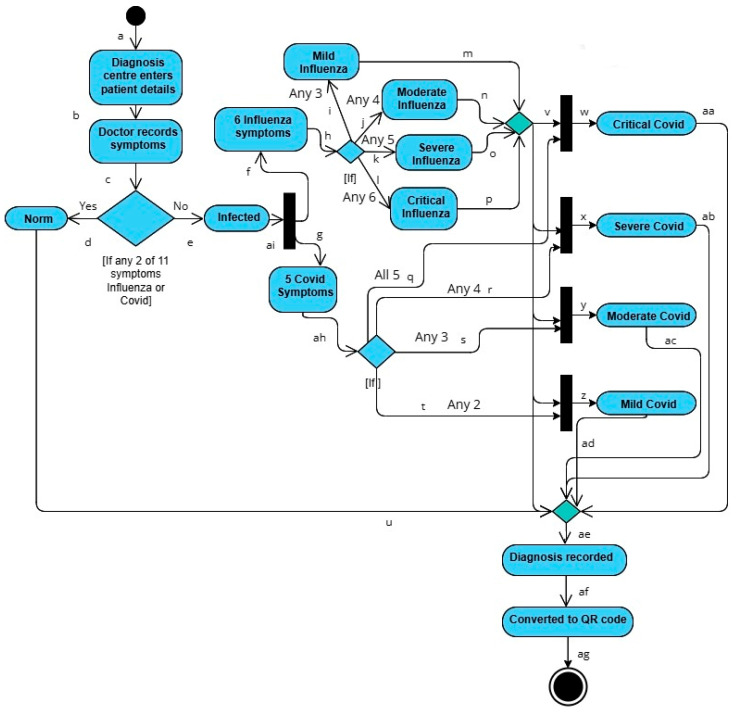
Activity diagram of the proposed grammar. The letters (a, b, c, d, etc.) provided as link labels are unrelated to the terminals of the grammar proposed and have been used to facilitate the attribute-based representation of this activity diagram.

**Figure 8 bioengineering-11-01265-f008:**
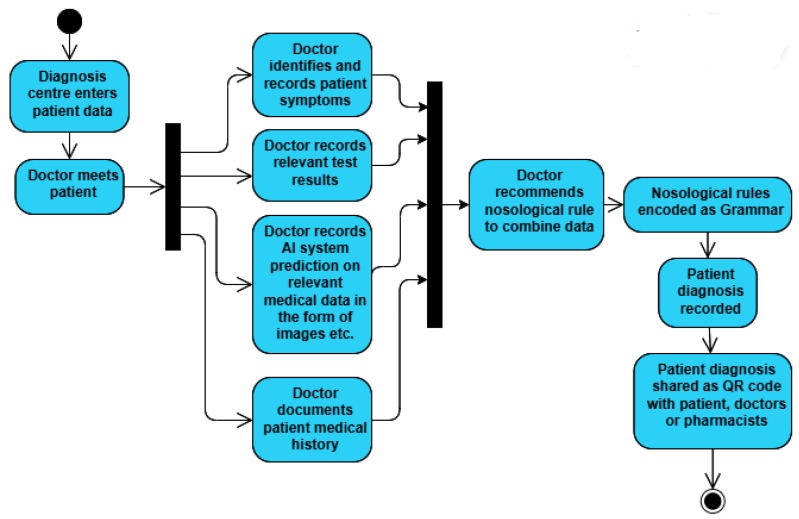
Activity diagram representing the extension of the proposed grammar for different diseases, incorporating the relevant test results, AI support, and patient’s medical history. The nosological rule that doctors have for using all the relevant information for diagnoses can be encoded as grammar and shared seamlessly among authorized personnel.

**Table 1 bioengineering-11-01265-t001:** Comparative study of different research studies pertaining to data collection, data sharing, and storage methods used for medical data.

Sl No.	Author	Research Problem	Research Details			
			Data Collection Type	Data Sharing	Data Storage	Workflow
1	Lynch et al. [64]	How to store and securely share patient genome data with pharmaceutical companies	Patient genomic data	Genomic data as secondary data	Cloud	NA
2	V. Santhana Marichamy, V. Natarajan [65]	How to manage and share electronic medical records in big data field	Patient’s medical records	Electronic medical records	Cloud [Pharmaceutical Companies]	Using blockchain and hadoop file sharing system. Blockchain developed using bitwise cryptographic hash generator.
3	H. Wu, X. Liu, W. Ou [66]	How to share medical data to the cloud platform for storage	Medical records	Medical records of patients	Cloud and data sharing between medical devices	Blockchain and multi-access edge computing (MEC).
4	Tao Zhang, Jian Shen, Chin Feng Lai, Sai Ji, Yongjun Ren [67]	How to develop multi-server assisted secure healthcare data sharing systems	Healthcare data records	Patient’s healthcare data	Public Server	Attribute-based encryption (ABE).
5	Swetha Gadde, J. Amutharaj, S. Usha [68]	How to develop a secure model to protect the isolation of medical data in the cloud	Medical data	Patient’s medical data	Cloud	Hybrid cryptography [S-box-based Advanced Encryption Standard (IRS-AES) with Runge–Kutta Optimization (RKO) algorithm]
6	Luis B. Elvas Carlos Serrão Joao C. Ferreira [69]	How to share secure heath information data	Healthcare data	Patient’s healthcare data	Cloud	Blockchain with design science research methodology (DSRM).
7	G. Muneeswari et al. [70]	How to store and securely share privacy-preserving medical data	Privacy-preserving medical data	Patients’ medical records	Cloud	Self-diagnosis platform (SDP) using IOT
8	Zigeng Fang et al. (review Article) [71]	Effective and secure management of healthcare system and coordinate all resources and data securely.	Healthcare data	Patient’s medical records	Cloud	A mixed-method approach was implemented to review and assess the existing efforts of open data city for healthcare through latitudinal and longitudinal analyses
9	Rahimi, M, Navimipour, NJ, Hosseinzadeh, M, Moattar, MH, Darwesh [72]	How to develop cloud-based healthcare services [review article]	Healthcare data	Medical records	Cloud	Clustering technique to explore several research papers
10	Mohiyuddin, A., Javed, A.R., Chakraborty, C. et al. [73]	Secure cloud storage for medical IoT data	Medical IoT data	Medical data of disease-free and diseased individuals	Cloud	Adaptive neuro fuzzy system
11	Jung-San Lee, Chit Jie Chew, Jo-Yun Liu, Ying-Chin Chen, Kuo-Yu Tsai [74]	Medical data sharing and privacy preserving	Medical data	Patients’ medical data	Cloud	Blockchain and smart contract
12	Jayapriya Jayabalan, N. Jeyanthi [75]	Healthcare data security and privacy	Healthcare data	Patients’ medical data	Cloud	They applied scalable blockchain with inter planetary file system (IPFS) for secure storage, sharing and retrieval of electronic health records (EHR) in healthcare management system
13	Suruchi Singh et al. [76]	Efficient data management in healthcare system	Healthcare data	Patients’ medical data	Cloud	Blockchain
14	Sun, Z., Han, D., Li, D. et al. [77]	Secure storage scheme for medical information	Medical data	Patients’ medical data	Cloud	Blockchain
15	L. Tan, K. Yu, N. Shi, C. Yang, W. Wei and H. Lu [78]	Secure and privacy-preserving data sharing for COVID-19 medical records	COVID-19 electric medical records (CEMRs)	Patient’s COVID-19 medical records	Cloud	Blockchain
16	Duo Zhang et.al. [79]	Secure and privacy-preserving medical data sharing	Medical data sharing	Patient’s medical data	Cloud	Quorum consortium blockchain
17	Lianhai Wang et al. [80]	Medical data sharing scheme for privacy preserving	Medical data sharing	Patient’s medical data	Cloud	Blockchain and smart contact storage
18	Yingwen Chen, Linghang Meng, Huan Zhou, and Guangtao Xue [81]	Medical data sharing mechanism with attribute-based access control and privacy protection	Medical data sharing mechanism	Medical data	Cloud	Blockchain
19	Scheibner J, Raisaro JL, Troncoso-Pastoriza JR, Ienca M, Fellay J, Vayena E, Hubaux JP. [82]	Revolutionizing medical data sharing	Medical data sharing mechanism	Medical data	Cloud	Homomorphic encryption and secure multiparty computation (defined together as multiparty homomorphic encryption)
20	Zarour M et al. [83]	Ensuring data integrity of healthcare information	Healthcare information	Patient’s healthcare information	Cloud	Review article
21	Zeng Chen, Weidong Xu, Bingtao Wang, Hua Yu [84]	preserving and sharing system for medical data privacy	Medical data sharing	Patient’s medical records	Cloud	Blockchain
22	Haddad, Alaa & Habaebi, Mohamed & Islam, Md. Rafiqul & Ahmad Zabidi, Suriza. [85]	Healthcare medical records management system with sharing control	Healthcare medical record management	Patient’s medical records	Cloud	Blockchain
23	Huang, Haiping et al. [86]	Privacy preserving and secure sharing of medical data	Secure sharing of medical data	Patient’s medical history	Cloud	Blockchain
24	Bharath H Rahul N, Shylash S, Vinny Pious [87]	Patient data management	Medical data	Patients’ medical history	Cloud	Blockchain
25	Liang Huang, Hyung-Hyo Lee [88]	Medical data privacy protection scheme	Medical data	Patients’ medical data	Cloud	Blockchain and cloud computing
26	Prokofieva, Maria & Miah, Shah [89].	Medical data security and secure sharing	Medical data sharing	Patient’s medical history	Cloud	Blockchain and smart contact
27	X. Liu, Z. Wang, C. Jin, F. Li and G. Li [90]	Medical data sharing and protection scheme	Patients medical data sharing	Electronic health record (EHR)	Cloud	Blockchain
28	Hao Jin, Yan Luo, Peilong Li and Jomol Mathew (review paper) [91]	Secure and privacy-preserving medical data sharing	Sharing medical data	Patient’s medical records	Cloud	The author classifies the research articles into permissionless blockchain-based approaches and permissioned blockchain-based approaches and analyzes their advantages and disadvantages
29	Chen, Wanghu & Mu, Yuxiang & Liang, Xiaoyan & Gao, Yaqiong. [92]	Medical data sharing model	Electronic medical data	Patients’ medical data	Cloud	Blockchain
30	Yan Luo Hao Jin Peilong Li [93]	Secure clinical data sharing and management	Medical data sharing	Patients’ medical records	Cloud	Blockchain
31	Dubovitskaya, Alevtina et al. [94]	Secure and trustable electronic medical records sharing	Electronic medical records (EMRs)	Patient’s data	Cloud	Secure and trustable EMR data management and sharing system using blockchain
32	X. Zheng, R. R. Mukkamala, R. Vatrapu and J. Ordieres-Mere [95]	Personal health data sharing system	Personal health management data	Patient healthcare data	Cloud	Blockchain
33	Q. Xia, E. B. Sifah, K. O. Asamoah, J. Gao, X. Du and M. Guizani [96]	MeDShare: trustless medical data sharing among cloud service providers	Trustless medical data sharing	Medical records	Cloud	Blockchain
34	Qinlong Huang, Licheng Wang, Yixian Yang [97]	Secure and privacy-preserving data sharing and collaboration in mobile healthcare social networks of smart cities	Electronic medical data	Human medical data	Cloud	Mobile healthcare social networks (MHSN)
35	Chin-Ling Chen; Jin-Xin Hu; Chun-Long Fan; Kun-hao Wang [98]	Secure medical data sharing system	Electronic medical data	Patient’s medical records	Cloud	biometric fingerprint feature and digital signature authentication for secure data storage
36	Asaph Azaria; Ariel Ekblaw; Thiago Vieira; Andrew Lippman [99]	Medical data access and permission management	Electronic medical records (EMRs)	Patient’s medical records	Cloud	Blockchain
37	Ahmed Lounis, Abdelkrim Hadjidj, Abdelmadjid Bouabdallah, Yacine Challal [100]	Healing on the cloud: secure cloud architecture for medical data	electronic medical records (EMRs)	Patient’s medical records	Cloud (healthcare providers)	Wireless sensor networks (WSN)
38	Yasmina Bensitel; Rahal Romadi [101]	Secure data storage in the cloud	Patient’s medical records	Patient’s medical records	Cloud	Homomorphic encryption
39	R. Karakış, İ. Güler, İ. Çapraz, E. Bilir [102]	Medical data security and management	Medical data (EEG) is selected as hidden data, and (MR) images are also used as the cover image	Patient’s medical records	Cloud	Fuzzy logic-based image steganography method
40	Ali Al-Haj, Gheith Abandah, Noor Hussein [103]	Secured medical image transmission	Medical data (DICOM image)	Patients’ medical records	Cloud	Cryptography-based algorithms
41	Chin-Ling Chen, Tsai-Tung Yang, Tzay-Farn Shih [104]	Secure medical data exchange protocol	Electronic medical records	Patients’ medical data	Cloud	Medical data exchange protocol, based on cloud environment
42	Soufiene Ben Othman; Abdullah Ali Bahattab; Abdelbasset Trad; Habib Youssef [105]	Secure medical data transmission protocol	Electronic medical records	Patients’ medical data	Cloud	Wireless sensor networks
43	Krzysztof Czuszynski; Jacek Ruminski [106]	Secure medical data exchange	Electronic medical records	Patients’ medical data	Cloud	Bar-codes and QR codes using Javascript
44	Mohamed Boussif, Noureddine Aloui, Adnene Cherif [107]	Secured cloud computing for medical data	Patient’s (health report and medical imaging)	Patients’ medical data	Cloud	Digital watermarking with Rashberry Pi

**Table 2 bioengineering-11-01265-t002:** An example of Top-Down and Bottom-Up parsers.

Grammar Rules	Parser Tree Generator							Output String
S → A A → A + B|B|a B → a*B|b	S→	A→	A	+	B			Top-Down Parser: a + a*b
			a	+	B			
					A	*	B	
							b	
	A	*	a	+	b		Bottom-Up Parser: a + a*b → S (Satisfied)	
	A	*	a	+	b			
	S	*	a	+	b			
			A	+	b			
			S	+	b			
					B			
					A			
					S			

**Table 3 bioengineering-11-01265-t003:** The diseases’ details [CDC Link in Appendix A].

Name	Symptoms	Non-Terminal Symbol	Terminal Symbol
Influenza	Fever	F	f
	Cough	C	c
	Body Ache	B	b
	Head Ache	H	h
	Sore Throat	T	t
	Runny Nose	R	r
Coronavirus	Trouble Breathing	D	d
	Fatigue	$F_{1}$	m
	Loss of Taste	L	l
	Vomiting	V	v
	Diarrhea	$D_{1}$	e

**Table 4 bioengineering-11-01265-t004:** Disease symptom criteria.

Type	No of Symptoms of Satisfied Condition
Normal Condition (NORM)	Any Two Symptoms
Mild Influenza (I1)	Any Three Symptoms of Influenza
Moderate Influenza (I2)	Any Four Symptoms of Influenza
Severe Influenza (I3)	Any Five Symptoms of Influenza
Critical Influenza (Need Immediate Medical Attention) (I)	All Six Symptoms of Influenza
Mild COVID-19 (C1)	Any Type of Influenza + Any Two Symptoms of COVID-19
Moderate COVID-19 (C2)	Any Type of Influenza + Any Three Symptoms of COVID-19
Severe COVID-19 (Need Medical Attention (C3)	Any Type of Influenza + Any Four Symptoms of COVID-19
Critical COVID-19 (Need Immediate Medical Attention) (COV)	Any Type of Influenza + All Symptoms of COVID-19

**Table 5 bioengineering-11-01265-t005:** Parse table for the unknown string, x = fchbtr (critical influenza found).

S					
I					
I_3_	I_3_				
I_2_	I_2_	I_2_			
I_1_	I_1_	I_1_	I_1_		
NORM	NORM	NORM	NORM	NORM	
F	C	H	B	T	R
**f**	**c**	**h**	**b**	**t**	**r**

**Table 6 bioengineering-11-01265-t006:** Parse table for the unknown string x = fchbtrdmlve (critical coronavirus found).

S										
COV										
C_4_	C_4_									
C_3_	C_3_	C_3_								
C_2_	C_2_	C_2_	C_2_							
C_1_	C_1_	C_1_	C_1_	C_1_						
I_4_	I_4_	I_4_	I_4_	I_4_	I_4_					
I_3_	I_3_	I_3_	I_3_	I_3_	I_3_	I_3_				
I_2_	I_2_	I_2_	I_2_	I_2_	I_2_	I_2_	I_2_			
I_1_	I_1_	I_1_	I_1_	I_1_	I_1_	I_1_	I_1_	I_1_		
NORM	NORM	NORM	NORM	NORM	NORM	NORM	NORM	NORM	NORM	
F	C	H	B	T	R	D	F_1_	L	V	D_1_
**f**	**c**	**h**	**B**	**t**	**r**	**d**	**m**	**l**	**v**	**e**

**Table 7 bioengineering-11-01265-t007:** Attribute-based representation of the activity diagram (Figure 7), based on the link labeling in Figure 7. In visual languages, the attaching points of a visual symbol (say, v) is numbered and represented by an array ap[1] to ap[n]. So, the value for ap[i] is the unique label assigned to the attaching point i of the visual symbol v (details available in Costagliola et al., 2004 [119], Costagliola et al., 2006) [139].

Name	ap[1]	ap[2]	ap[3]	ap[4]	ap[5]	ap[6]	ap[7]
Start	{a}						
Diagnosis center enters patient details	{a}	{b}					
Doctor records symptoms	{b}	{c}					
If conditional	{c}	{d}	{e}				
Norm	{d}	{u}					
Infected	{e}	{ai}					
Sync1	{ai}	{f,g}					
6 influenza symptoms	{f}	{h}					
If conditional	{h}	{i}	{j}	{k}	{l}		
Mux1	{m}	{n}	{o}	{p}	{v}		
5 COVID-19 symptoms	{g}	{ah}					
If conditional	{ah}	{q}	{r}	{s}	{t}		
Sync2	{v,q}	{w}					
Sync3	{v,r}	{x}					
Sync4	{v,s}	{y}					
Sync5	{v,t}	{z}					
Critical COVID-19	{w}	{aa}					
Severe COVID-19	{x}	{ab}					
Moderate COVID-19	{y}	{ac}					
Mild COVID-19	{z}	{ad}					
Mux2	{u}	{v}	{aa}	{ab}	{ac}	{ad}	{ae}
Diagnosis recorded	{ae}	{af}					
Converted to QR code	{af}	{ag}					
Halt	{ag}						

## Data Availability

Data are contained within the article and Appendix A.

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
