# Peer review of "A Novel Grammar-Based Approach for Patients’ Symptom and Disease Diagnosis Information Dissemination to Maintain Confidentiality and Information Integrity"

_bioengineering, 2024, doi:10.3390/bioengineering11121265_

Round 1
Reviewer 1 Report
Comments and Suggestions for Authors
The paper is a very good one but can be improved.
In your abstract you made mention of disease several times but it is not reflecting in the title of the paper. I will advise that it should be incorporated in the title.
Figure 2 is wrongly placed; I believe it should be under the methodology section.
line 512 should be rephrased because I don't understand the statement in its present form.
Comments on the Quality of English LanguageMinor grammatical errors.
Author Response
Reviewer 1: Comments & Suggestions
Observation 1: The English could be improved to more clearly express the research.
Thank you for your observation. In view of your observation, we have re-written sections in the manuscript (highlighted in yellow) and also corrected grammatical errors to the best of our ability to make our explanation of the research clearer and easily accessible.
Comment 1: The paper is very good one but can be improved.
Thank you for your comment. Your comment is indeed humbling. We are indeed grateful. As outlined in our explanation for Observation 1, we have re-written sections in the manuscript (highlighted in yellow) and also corrected grammatical errors to the best of our ability to improve the manuscript.
Comment 2: In your abstract you made mention of disease several times but it is not reflecting in the title of the paper I will advise that it should be Incorporated in the title
Thank you for your comment. In accordance with your comment, we have now amended the title of the paper from
‘A Novel Grammar Based Approach for Patient Information Dissemination to Maintain Confidentiality and Information Integrity’ to
‘A Novel Grammar Based Approach for Patient Symptomatic and Disease Diagnosis Information Dissemination to Maintain Confidentiality and Information Integrity’
Comment 3: Figure 2 is wrongly placed. I believe it should be under the methodology section
Thank you for pointing this out. In view of your suggestion, we have now moved Figure 2 from the ‘Introduction’ (Section 1) to ‘Experiment Results, Applications and Discussions’ (Section 4). We felt that Figure 2 is more focused on the application of the proposed method, hence it was moved to Section 4.
Comment 4: line 512 should be rephrased because I don't understand the statement in its present form
Thank you for pointing this out. As per your comment, we have replaced the following statement in line 512:
‘Use of automation as a support to medical practitioners is the need of the hour.’ with
‘Ease of relevant secure accessibility of sensitive data with limited resource to support medical practitioners for information sharing would help strengthen the present medical infrastructure.’
Comment 5: Minor grammatical errors
Thank you for your comment. In light of your comment, we have reviewed the manuscript and corrected grammatical errors (highlighted in yellow) to the best of our ability.
For example, the sentence
‘The two disease that is focused in this research article is Influenza and Corona Virus infections. (line 209)’ has been replaced by
‘The two diseases that are in focus in this research article are Influenza and Corona Virus infections.’
[N.B. : The line numbers indicated are in view of the original manuscript that was submitted to the journal. The Highlighted sections are available in the Word document submitted in Supplementary File(s)]

Reviewer 2 Report
Comments and Suggestions for Authors
In general the article is well organized and the medical application of the novel logical grammar based on the Chomsky Normal Form and the Cocke-Younger-Kasami algorthm is also very interesting.
However, although the English can be understood in general it could be greatly improved, because in many sentences there is a lack of verbs, adverbs and pronouns that makes it incorrect.
Just to mention a few of cases:
- in the Abstract at the very first sentence "Disease prediction ... is now established research."
I could introduce "an" before "research". The same before "important field" in line 42.
- I would change "literatures" by "literature"
- I also would add "a" before "mobile application"
- I would add "than" before "affects" at line 72.
- delete "was" at the end of line 82.
- correct "Forrest" at line 173.
- correct the name of the reference [48] at line 207.
- add "it" before "is shown" at line 243.
- it could be better to center the captions of the figures and tables.
- add "a" before "limited number of specialty hospitals" at line 272.
- add "is" before "defined by the language" at line 328.
- the sentences at lines 351 and 352 are not clear or understandable.
- include an space before "and" at line 365.
- at line 430 change "illustrate" by "illustrated".
- at lines 432-433 include "is" before "present" and "accepted" and include "that" before "the unknown".
- add "of" just before "conditions" at line 462.
- "Human factor" at line 477.
- "minimalist technology" at line 479.
Author Response
Reviewer 2: Comments & Suggestions
Observation 1: In general, the article is well organized and the medical application of the novel logical grammar based on the Chomsky Normal Form and the Cocke-Younger-Kasarni algorthm is also very interesting.
Thank you for your comment. Your comment is indeed encouraging. We are grateful to you for your kind words.
Comments 1: However, although the English can be understood in general it could be greatly improved because in many sentences there is a lack of verbs, adverbs and pronouns that makes it incorrect.
Thank you for your comment. In light of your observation, we have re-written sections of the manuscript (highlighted in yellow) in our pursuit to correct grammatical errors and improve the overall readability of the paper.
Comments 2: In the Abstract at the very first sentence “Disease prediction is now established research” I could introduce an before "research" The same before “important field- in “
Thank you for your comment. In accordance with your suggestion, we have now replaced the following sentences:
‘Disease prediction using computer-based methods is now established research’ and
‘Medical research is important field of science and technology that contributes towards disease diagnosis, prevention and cure.’ (line 42)
with
‘Disease prediction using computer-based methods is now an established research’ and
‘Medical research is an important field of science and technology that contributes towards disease diagnosis, prevention and cure.’ (line 42) respectively.
Comments 3: line 42 - I would change 'literatures" by literature-
Thank you for your comment. In accordance with your suggestion, we have now replaced ‘literatures’ with ‘literature’ in line 16.
Comments 4: I also would add "a" before "mobile application"
Thank you for your comment. In view of your comment, we have replaced the sentence,
‘The code can be stored in mobile application in a secured manner and can be scanned.’ (line 22) with
‘The code can be stored in a mobile application in a secured manner and can be scanned.’
Comments 5: I would add “than” before "affects" at line 72
Thank you for your comment. In our effort to re-write sections of the paper, we have rephrased the following sentence,
‘This is a seasonal disease affects all age groups with minor to severe symptoms.’ (line 71-72)
with
‘This seasonal disease affects all age groups with minor to severe symptoms.’
Comments 6: delete "was" at the end of line 82
Thank you for your comment. The following sentence,
‘The virus is known to cause pandemic with at least three major pandemic was reported in the last century [8] and one caused by influenza A (H1N1) virus some years back.’ (line 84 -86)
with
‘At least three major pandemics were caused by virus were reported in the last century [8]. One of these pandemics was caused by influenza A (H1N1) virus.’
Comments 7: correct "Forrest' at line 173
Thank you for pointing out this typographic error. We have now corrected it.
Comments 8: correct the name of the reference [481 at line 207
Thank you for pointing this out. We have now corrected the error in the manuscript.
Comments 9: add "it" before "is shown" at line 243
Thank you for your comment. We have now updated the following sentence,
‘An overlay visualization of is shown in Figure 3 depicting the associations of different keywords used in the studies mentioned in this section.’
with
‘An overlay visualization of it is shown in Figure 3 depicting the associations of different keywords used in the studies mentioned in this section.’
Comments 10: It could be better to center the captions of the figures and tables
Thank you for your suggestion. The captions of the figures and tables have now been centered.
Comments 11: add "a" before "limited number of specialty hospitals at line 272
Thank you for your comment. In our effort to improve the readability of the manuscript we have now removed the sentence under consideration.
Comments 12: Add "is" before 'defined by the language" at line 328
Thank you for your comment. We have added ‘is’ before 'defined by the language’ (highlighted in yellow).
Comments 13: The sentences at lines 351 and 352 are not clear or understandable
Thank you for pointing this out. The following two sentences,
‘The grammar contains G no of rules.
w is the n length string to be parsed.’
have been replaced with
‘The grammar G contains no rules.
w is the string of length n to be parsed.’
Comments 14: Include an space before "and" at line 365
Thank you for your comment. A space has been added between cell(i,k) and ‘and’.
Comment 15: at line 430 change "illustrate" by 'Illustrated'
Thank you for pointing this out. We have replaced ‘illustrate’ with ‘illustrated’ in line 430.
Comment 16: at lines 432-433 include "is" before "present" and "accepted" and include "that" before "the unknown"
Thank you for your comment. The following sentence,
‘If the start symbol S present in that location that means the unknown disease string accepted by the grammar and satisfy the rules.’
has been replaced with
‘If the start symbol S is present in that location that means that the unknown disease string is accepted by the grammar and it satisfies the rules.’
Comment 17: add of just before "conditions'. at line 462
Thank you for your comments. We are unsure about the error in the following sentence,
‘There are myriad conditions that may lead to a disease condition.’
Hence it was not updated.
Comment 18: "Human factor" at line 477
Thank you for your comment. We have now replaced, ‘humane factor’ with ‘human factor’.
Comment 19: "minimalist technology" at line 479
Thank you for pointing this out. We have now replaced ‘minimalistic technology’ with ‘minimalist technology’.
[N.B. : The line numbers indicated are in view of the original manuscript that was submitted to the journal. The Highlighted sections are available in the Word document submitted in Supplementary File(s).]
Reviewer 3 Report
Comments and Suggestions for Authors
The paper proposes an approach to make desease diagnosis, which relies on grammars rather than machine learning, with the potential advantage
of requiring limited computation time.
al langu
Authors belief is that artificial intelligence is not the solution for everything, especially in the field of desease diagnosis, where they
believe that it should be the human defined rules that are helpful. Thus, the method they propose seeks to use the perspective of the doctors
in disease diagnosis and encode them in the form of syntactic grammars.
The paper has many important problems and cannot be accepted for publication as it is now. In what follows I list the main (serious) problems
of authors' proposal:
- First of all, the motivation. It is not true that human centered (doctors) diagnosis are not based on artificial intelligence. There is a
wide literature in the 80's (40 years ago) concerning rule based systems for diagnosis, among which the Mycin system, not mentioned by the authors. One of the main problems with such systems, which I think will also affect authors' proposal, is that they are not able to update
their knowledge without a complex and expensive knowledge engineering activity, which made such systems be abandoned by industry. This is the reason why modern systems rely on machine learning, because they should somehow be able to update their knowledge when underlying data change:
- Although this is not properly a Computer Science based venue, but one focused on Bioengineering, from a Computer Scientist point of view the
paper does not propose anything new. Authors just use a well known grammar for a given application domain. In doing this, authors completely
ignore a wide literature on visual language grammars that have abounded in the literature between the '80s and the '90s. Such grammars are
much more powerful than context-free grammars used for textual languages, since they enable to model not only symbols, but any kind of relation
between them, and not just left-right relations as in grammars for textual languages. Thus, such grammars would enable modeling many more
specific situations concerning diagnosis, such as different temporal relations between sympthoms, and so on, which makes me think there is
something that can be modeled by this grammars, but not in authors' proposal. To this end, authors do not make any formal analysis on the
completeness of their approach w.r.t. to the problem they try to model. Besides, for some of visual language grammars there are also automatic
parser generation tools YACC-Like, which make the generation of a parser quite straightforward. An example is provided in the following paper:
G. Costagliola, et. al, “Visual Language Implementation Through Standard Compiler Compilers”, International Journal of Visual Languages and Computing, Volume 18, Issue 2, April 2007, pp. 165-226.
In that paper authors can also find a rich literature on visual language grammars in the list of references.
- Visual language grammars have also been used to generate visual language based systems to specify complex protection policies in a user friendly way, which is one of the problems mentioned by the authors at page 4 of their paper. To this end, for future revisions of their
paper authors might want to take a look at the following paper and its references:
M. Giordano, et. al, “Visual Computer-Managed Security: A Framework to Support Access Control in Enterprise Applications”, IEEE Software, Vol. 30, No. 5, 2013, pp. 62-69.
- Continuing the discussion on the insufficient bibliography survey, since the paper also mentions the problem of Alzheimer’s Disease, authors missed some relevant recent paper, like for instance
G. De Gregorio, et. al., A Multi Classifier Approach for Supporting Alzheimer’s Diagnosis Based on Handwriting Analysis, International Workshop on Artificial Intelligence for Healthcare Applications (AIHA2020).
The reference list of such paper might also be useful to authors for future revisions and enrichments of their paper.
- The paper is not well organized. Authors do not properly state their contribution in the introduction section. Although this section is
extremely long, at the end of it no paragraph is provided with a description of paper sections.
MINOR POINTS:
- Page 3, line 2, I think between the words "unavailability" and "an" the word "of" should be inserted
- Page 6, line 1, shouldn't it be "deseases" instead of "desease"?
- Page 6, line 5, shouldn't it be "organizations" instead of "organization"?
- Page 6, line 11, I think a comma should be inserted after the word "predict"
- Page 6, line 16, this line ends with an orphan
- Page 14, line 10, the sentence "The application of grammar with definite set of production rules on the obtained features 302 for classification" should be reworked
- Page 14, line marked 326, I think something is missing between the words "string" and "defined"
- Page 15, table 2, the section on the bottom up parsing uses a notation that is not clear. In the references I provided above much more effective notations are used to show the evolution of a bottom up parsing process
- Title section 4.2, maybe should be "Discussion"?
Author Response
Reviewer 3: Comments & Suggestions
Observation 1: The paper proposes an approach to make disease diagnosis, which relies on grammars rather than machine leaming, with the potential advantage of requiring limited computation time.
The paper does propose an approach to grammars for recording symptomatic conditions and outcomes of machine learning algorithms for diagnosis to provide effective, limited resource based responsible sharing of sensitive medical information. Apologies for the gap in communication.
Observation 2: Authors belief is that artificial intelligence is not the solution for everything, especially in the field of disease diagnosis, where they believe that it should be the human defined rules that are helpful. Thus, the method they propose seeks to use the perspective of the doctors in disease diagnosis and encode them in the form of syntactic grammars.
We believe that the artificial intelligence-based toolkits can support diagnosis/decision making by registered medical practitioners and cannot be considered as replacement for the medical practitioners. The medical practitioners use nosological rules in coherence with predicted outcomes by validated artificial intelligence or machine learning systems to arrive at a diagnosis.
We believe, artificial intelligence systems are still limited in their ability to encompass previous medical history of a patient in coherence with physical examination reports at a given point of time to arrive at a diagnosis.
Comment 1 (Major Issue): First of all, the motivation. It is not true that human-centred (doctors) diagnosis is not based on artificial intelligence. There is a wide literature in the 80's (40 years ago) concerning rule-based systems for diagnosis, among which the “Mycin system”, not mentioned by the authors. One of the main problems with such systems, which I think will also affect authors’ proposal, is that they are not able to update their knowledge without a complex and expensive knowledge engineering activity, which made such systems be abandoned by industry. This is the reason why modem systems rely on machine learning, because they should somehow be able to update their knowledge when underlying data change:
Our apologies for the gap in communication and the unwanted confusion. We do believe that artificially intelligent systems do play a critical role in supporting diagnosis/decision making by doctors. However, we do believe that they can only support decision making by doctors. This is because unlike artificially intelligent systems doctors use nosological rules, past medical history of a patient along with their examination or imaging reports for disease diagnosis.
We do agree nosological rule-based systems unlike machine learning based systems cannot automatically learn in view of new data. But no amount of learning can replace the skill of the doctors at analysing past medical history of patients, disease symptoms and analysis of medical examination/ imaging reports by a validated artificially intelligent system to responsibly arrive at a diagnosis.
It might be worth mentioning that supervised or semi-supervised artificially intelligent systems start their learning based on data annotated by medical practitioners.
We are proposing a grammar-based method for limited resource-based encoding of symptomatic, examination and diagnosis data which can be shared responsibly and securely among medical practitioners and pharmacologists. However, the extendable production rules for the grammar should be based on symptomatic, test results, medical history based and technologically supported rules used by medical practitioners for diagnosis.
Comment 2 (Major Issue): Although this is not properly a Computer Science based venue, but one focused on Bioengineering, from a Computer Scientist point of view the paper does not propose anything new. Authors just use a well-known grammar for a given application domain.
Thank you for your comment. The novelty of the work lies in the use of the well-known grammar for secure yet seamless coding and thereby sharing of diseases diagnosis information between doctors, hospitals and pharmacists in a resource effective manner. With the growing influx of pandemics, need for environmental conservation, we believe such coding and resource limited, paperless, secure sharing of information is a concept that will support human community wellbeing and is the need of the present-day world. As outlined by the reviewer, the novelty of the work lies in its application in effective information storage and secure sharing in the medical domain.
Comment 3 (Major Issue): In doing this, authors completely ignore a wide literature on visual language grammars that have abounded in the literature between the ‘80s and the ‘90s. Such grammars are much more powerful than context-free grammars used for textual languages, since they enable to model not only symbols, but any kind of relation between them, and not just left-right relations as in grammars for textual languages. Thus, such grammars would enable modeling many more specific situations concerning diagnosis, such as different temporal relations between symptoms, and so on, which makes me think there is something that can be modeled by these grammars, but not in authors’ proposal. To this end, authors do not make any formal analysis on the completeness of their approach w.r.t. the problem they try to model.
Thank you for your valuable comment. This comment was indeed helpful. Our initial idea was to have a simple grammar-based representation to showcase our concept. Again, complex parsing algorithms are liable to have exponential time and space complexity which defeats the resource limited approach that we have put forth in the paper.
However, we do agree with the reviewer that a graphical or visual grammar has more scope and flexibility to represent different temporal relationships between symbols.
Hence, we have now incorporated activity diagram (Figure 7) in the main document and an attribute-based representation of the activity diagram of our proposed grammar (Table 7). We do believe, addition of this Table and Figure has made our work more complete and comprehendible.
Once again, thank you for the insight. It was indeed helpful.
Comment 4 (Major Issue): Besides, for some of visual language grammars there are also automatic parser generation tools YACC-Like, which make the generation of a parser quite straightforward. An example is provided in the following paper:
- Costaglola, et. al, “Visual Language Implementation Through Standard Compiler Compilers’, Intemational Journal of Visual Languages and Computing, Volume 18, Issue 2, April 2007, pp. 165-226.
Thank you for pointing this out. We have gone through the paper suggested and also done our own research to get a better understanding of the concepts proposed. We shall reflect on this when documenting other similar manuscripts.
Comment 5 (Major Issue): In that paper authors can also find a rich literature on visual language grammars in the list of references. - Visual language grammars have also been used to generate visual language-based systems to specify complex protection policies in a user-friendly way, which is one of the problems. mentioned by the authors at page 4 of their paper. To this end, for future revisions of their paper authors might want to take a look at the following paper and its references:
- Costagliola, et. al, “Visual Language Implementation Through Standard Compiler Compilers’, International Journal of Visual Languages and Computing, Volume 18, Issue 2, April 2007, pp. 165-226.
Thank you for your comment. As indicated in the last comment, we have read the paper and do believe it has been particularly helpful in making our work completer and more comprehensive. We have included aspects of visual language representation in our work (particularly in Figure 7 and Table 7).
Comment 6 (Major Issue): Continuing the discussion on the insufficient bibliography survey, since the paper also mentions the problem of Alzheimer's Disease, authors missed some relevant recent paper, like for instance
- De Gregorio, et. al, A Multi Classifier Approach for Supporting Alzheimer's Diagnosis Based on Handwriting Analysis, International Workshop on Artificial Intelligence for Healthcare Applications (AIHA2020).
The reference list of such paper might also be useful to authors for future revisions and enrichments of their paper.
Thank you for your suggestion. The paper has been included in the in-text and final reference list for the document (Reference no. 37 in Bibliography/Reference List).
Comment 7 (Major Issue): The paper is not well organized, Authors do not properly state their contribution in the introduction section. Although this section is extremely long, at the end of it no paragraph is provided with a description of paper sections.
Thank you for your comment. In accordance with your suggestion, we have included a couple of paragraphs at the end of ‘Introduction’ section. The first of the two paragraphs defines the contents of the different sections of the paper and the last paragraph outlines the novel contribution of the proposed work.
Comment 8 (Minor points): Page 3, line 2, I think between the words "unavailability" and "an" the word "or should be inserted
Thank you for your comment. We have replaced the following sentence,
‘Moreover, unavailability an inadequate medical infrastructure (even in economically & technologically advanced countries) was the primary reason for high mortality all across the world.’
with
‘Moreover, unavailable or an inadequate medical infrastructure (even in economically & technologically advanced countries) was the primary reason for high mortality all across the world.’
Comment 9 (Minor points): Page 6, line 1, shouldn't it be "deseases" instead of "desease"?
Thank you for pointing this out. The following sentence,
‘The two disease that is focused in this research article is Influenza and Corona Virus infections. (line 209)’ has been replaced by
‘The two diseases that are in focus in this research article are Influenza and Corona Virus infections.’
Comment 10 (Minor points): Page 6, line 5, shouldn't it be "organizations" instead of "organization"?
Thank you for pointing this out. We have now replaced ‘organization’ with ‘organizations’.
Comment 11 (Minor points): Page 6, line 11, I think a comma should be inserted after the word "predict"
Thank you for your comment. We have inserted a comma after predict.
Comment 12 (Minor points): Page 6, line 16, this line ends with an orphan
Thank you for your comment. Not sure what needs to be addressed here.
Comment 13 (Minor points): Page 14, line 10, the sentence "The application of grammar with definite set of production rules on the obtained features for classification" should be reworked
Thank you for pointing this out. The following sentence,
‘The application of grammar with definite set of production rules on the obtained features for classification.’
has been replaced by
‘The application used a grammar with definite set of production rules on the obtained features for classification.’
Comment 14 (Minor points): Page 14, line marked 326, I think something is missing between the words "string" and "defined"
Thank you for pointing this out. The following sentence,
‘As a result, a comprehensive examination of all derivations can be used to establish whether a string defined by the language.’
has been replaced by
‘As a result, a comprehensive examination of all derivations can be used to establish whether a string is defined by the language.’
Comment 15 (Minor points): Page 15, table 2, the section on the bottom-up parsing uses a notation that is not clear. In the references I provided above much more effective notations are used to show the evolution of a bottom-up parsing process
Thank you for your comment. However, given the limited time to undertake all suggested review work, the symbols haven’t been altered. However, in view of your recommendation we have included visual notations to represent our proposed grammar in the document (Figure 7 and Table 7 in the main document).
Comment 16 (Minor points): Title section 4.2, maybe should be "Discussion"?
Thank you for pointing this out. The Title section 4.2 ‘Discussions’ has now been replaced with ‘Discussion’.
[N.B. : The line numbers indicated are in view of the original manuscript that was submitted to the journal. The Highlighted sections are available in the Word document submitted in Supplementary File(s).]
Round 2
Reviewer 3 Report
Comments and Suggestions for Authors
Although authors have addressed many of my concerns regarding the literature, I think the paper still needs a major revision before it can be
considered for publication:
- I provided authors with several papers from the visual language bibliography, suggesting them to look into their bibliography to extract main
contributions in this area. Authors have only considered the papers I suggested, ignoring main literature of the area.
- Authors should better explain how they addressed my criticisms concerning their proposal. I understand that their position is against AI, reasonably
saying that AI alone is not sufficient without human intervention. But I don't see explained why the grammar based solution would solve these
problems.
I need these two remarks be thoroughly addressed in next revision.
Further proofreading is needed.
Author Response
Reviewer 3 : Comments & Suggestions
Observation 1 : Although authors have addressed many of my concerns regarding the literature, I think the paper still needs a major revision before it can be considered for publication:
O - Response 1 : Thank you for your kind observation. We do feel your comments have been helpful and will work towards addressing your concerns for improving the manuscript further.
Comment 1 : I provided authors with several papers from the visual language bibliography, suggesting them to look into their bibliography to extract main contributions in this area. Authors have only considered the papers I suggested, ignoring main literature of the area.
C - Response 1 : Thank you for your comment. We would like to bring to your notice that the time provided for making amends based on reviewer comments is tight and given this being term time it becomes a stretch to undertake extensive literature review. However, we have managed to include review of the main contributions in the visual language area in the revised version of our document. It is highlighted in yellow in the Supplementary material/document provided.
In the main document this is available in Page 18, Line number 477 to line number 507.
Comment 2 : Authors should better explain how they addressed my criticisms concerning their proposal. I understand that their position is against AI, reasonably saying that AI alone is not sufficient without human intervention. But I don't see explained why the grammar based solution would solve these problems.
C - Response 2 : Thank you for your comment. We do believe your comments have been helpful in refining or adding more clarity to our work. However, we disagree that we are against AI. Our claim as you mentioned is AI should be used to support decision making by registered medical professionals. At no point have we claimed that the grammar-based solution proposed will replace or solve the short-comings of AI. The novelty of our proposed work lies in the ability to code the nosological rules used by medical practitioners based on symptomatic data, outcomes of AI support systems, medical history of patients and relevant medical tests to diagnose and decide the disease that the patient might be suffering from. Now, this coded information by use of a grammar will particularly be light weight (i.e. less resource intensive) and securely portable and shareable using a QR code with patients and other authorized personnel seamlessly. Not a replacement for AI, but an extension of AI support systems to facilitate transparent, unambiguous, secure sharing of symptomatic data, medical history, AI outcomes, relevant medical tests that have been used by the doctor to reach a diagnosis.
We understand that Figure 5 in the document might seem counter intuitive, hence we have updated Figure 5. This is highlighted in yellow in the Supplementary material/document. Also, we have added some text and an Activity diagram (Figure 8) to clear our standpoint with how the proposed model can be used in recording of the parameters used by a medical practitioner to arrive at a diagnosis. This is highlighted in yellow in the Supplementary material/document.
In the main document this is available in Page 22-23, Line number 539 to line number 548.
Observation 2 : I need these two remarks be thoroughly addressed in next revision.
O - Response 2 : Once again thank you for your valuable comments. We feel we have done our best to address your concerns.
Comment 3 : Further proofreading is needed.
C-Response 3 : In view of your suggestion, we did proofread the document and corrected typographic errors. This is indicated in yellow in the Supplementary material/document.